# Nano-scale collinear multi-Q states driven by higher-order interactions

Mara Gutzeit[1], André Kubetzka [2], Soumyajyoti Haldar [1], Henning Pralow[1], Moritz A. Goerzen [1], Roland Wiesendanger[2], Stefan Heinze [1,3] & Kirsten von Bergmann [2] ✉

Complex magnetic order arises due to the competition of different interactions between the magnetic moments. Recently, there has been an increased interest in such states not only to unravel the fundamental physics involved, but also with regards to applications exploiting their unique interplay with moving electrons. Whereas it is the Dzyaloshinskii-Moriya interaction (DMI) that has attracted much attention because of its nature to induce non-collinear magnetic order including magnetic-field stabilized skyrmions, it is the frustration of exchange interactions that can drive magnetic order down to the nano-scale. On top of that, interactions between multiple spins can stabilize two-dimensional magnetic textures as zero-field ground states, known as multi-Q states. Here, we introduce a two-dimensional itinerant magnet with various competing atomic-scale magnetic phases. Using spin-polarized scanning tunneling microscopy we observe several zero-field uniaxial or hexagonal nano-scale magnetic states. First-principles calculations together with an atomistic spin model reveal that these states are stabilized by the interplay of frustrated exchange and higher-order interactions while the DMI is weak. Unexpectedly, it is found that not only non-collinear magnetic states arise, but that higher-order interactions can also lead to collinear nano-scale multi-Q states.

Multi-Q states are complex, often two-dimensionally periodic magnetic states, which are typically non-collinear and non-coplanar[1]. They have been discussed recently both in the context of exchange frustrated systems[1–5], as well as in the field of skyrmion lattices and related topological objects[6–8], and can exhibit exceptional transport properties such as a large anomalous or topological Hall effect[5,8,9]. Multi-Q states are superposition states, and their building blocks—the single-Q states—are one-dimensionally modulated magnetic states, i.e., spin spirals. Spin spirals can arise due to competing magnetic interactions, such as frustration of exchange interactions, or contributions from the Dzyaloshinskii–Moriya interaction (DMI). The interplay of these different interactions leads to a specific period in real space, or so-called $Q$-vector in reciprocal space.

Various magnetic interactions beyond pair-wise Heisenberg exchange or DMI have been proposed such as biquadratic or four-spin exchange[10–12], topological-chiral[13,14] or chiral multi-spin interactions[15–17]. Such higher-order terms can couple spin spirals to form multi-Q states[18,19]. A prominent example is the triple-Q state in a hexagonal magnetic monolayer, which was predicted 20 years ago for Mn/Cu(111)[18] and observed only recently in Mn/Re(0001)[20]. It is one of the few exact multi-Q states that is degenerate with its constituting single-Q states in the absence of higher-order terms, and it has the same magnetic moment at every site.

For a given length of a $Q$-vector and the respective symmetry of the system a plethora of multi-Q states can be constructed. However, most of these superposition states do not have a constant magnetic

[1]Institute of Theoretical Physics and Astrophysics, University of Kiel, Leibnizstrasse 15, 24098 Kiel, Germany. [2]Department of Physics, University of Hamburg, 20355 Hamburg, Germany. [3]Kiel Nano, Surface, and Interface Science (KiNSIS), University of Kiel, Kiel, Germany. ✉e-mail: kbergman@physnet.uni-hamburg.de

moment at every lattice site. Therefore, additional non-symmetry-equivalent $Q$-vectors such as higher harmonic $Q$-vectors or a ferromagnetic ($Q = 0$) contribution are often taken into account. Many of the recently predicted multi-Q states with different symmetry and topological charge[1–5] do not require higher-order interactions for their formation but they are instead triggered by applied magnetic fields. One special type of field-induced multi-Q state is the hexagonal skyrmion lattice[6,21,22], which contains three symmetry-equivalent spin spirals with unique rotational sense and type according to the system-specific DMI.

Spontaneous skyrmion lattices, that are stabilized at zero field by higher-order magnetic interactions arising between multiple spins, have been observed experimentally in an Fe monolayer on Ir(111)[7,23]. Further zero-field non-collinear multi-Q states have been theoretically proposed[24,25]. A recent study has reported the combination of higher-order exchange and magnetic field as origin for a square skyrmion lattice state in a centrosymmetric tetragonal magnet, in which the DMI vanishes[25–27]. However, the understanding which types of multi-Q states or skyrmion lattices are possible due to the interplay of frustrated pair-wise exchange, DMI, and higher-order exchange is still rather limited. Material systems studied both experimentally and by first-principles theory provide valuable insights but remain scarce.

Here, we show that Fe/Rh atomic bilayers on the Ir(111) surface are strongly frustrated two-dimensional magnetic systems with a large number of competing magnetic phases. Using spin-polarized scanning tunneling microscopy (SP-STM) we demonstrate that a change of the stacking of the Fe monolayer (ML) or the number of Rh layers leads to a variety of different uniaxial or hexagonal nano-scale magnetic spin textures in zero magnetic field. First-principles electronic structure theory is employed to reveal the origin of these different magnetic states. In particular the competition of frustrated Heisenberg exchange and higher-order interactions is the driving force for spontaneous single-Q or multi-Q state formation. We find that the recently proposed three-site four-spin interaction[12] plays a decisive role in the stabilization of hexagonal spin textures. Surprisingly—and in contrast

to previous two-dimensionally modulated multi-Q states—we find that in our system the higher-order interactions penalize non-collinear spin arrangements and induce collinear nano-scale magnetic order.

## Results

### Experimental results

Figure 1 shows an overview SP-STM image of a sample of Fe on Rh/Ir(111). Both Fe and Rh have sub-monolayer coverage. A detailed examination of the different exposed areas leads to the layer assignment as indicated by the colored dots (compare the side-view sketch below for the specific local configuration, and methods section for more details on the preparation and assignment of the layers and stackings). All of the labeled areas are pseudomorphic. Independent of the layer sequence and stacking, the Fe-ML areas on Rh show nano-scale superstructures, which originate from the magnetic texture. In the following we will take a closer look at the details of the magnetic states in the different Fe-MLs on single and double fcc-stacked layers of Rh (named Rh1 and Rh2) on Ir(111).

In Fig. 2 we concentrate on Fe/Rh1, and several sample areas with the two different Fe-stackings are indicated in the overview image of Fig. 2a. Figure 2b, c show constant-current data of these areas with the identical height scale of $\Delta h = 50$ pm from black to white. The magnetic superstructure we observe on fcc-Fe/Rh1 (Fig. 2b) is uniaxial and the stripes run parallel to the close-packed atomic rows. All three rotational domains are present and apparently the stripes prefer to run across the shorter side of elongated islands with a distance of about 1.18 nm, which corresponds roughly to 5 atomic rows. We find that the magnetic state is not in perfect registry with the atomic lattice and in addition to this incommensurability the stripes tend to show a faint substructure. The application of an external magnetic field (right side) does not change the appearance of the magnetic structure in fcc-Fe/Rh1. As the three rotational domains have similar magnetic corrugation amplitudes we can conclude based on symmetry arguments that the observed magnetic contrast dominantly originates from out-of-plane sample magnetization components. Two different spin

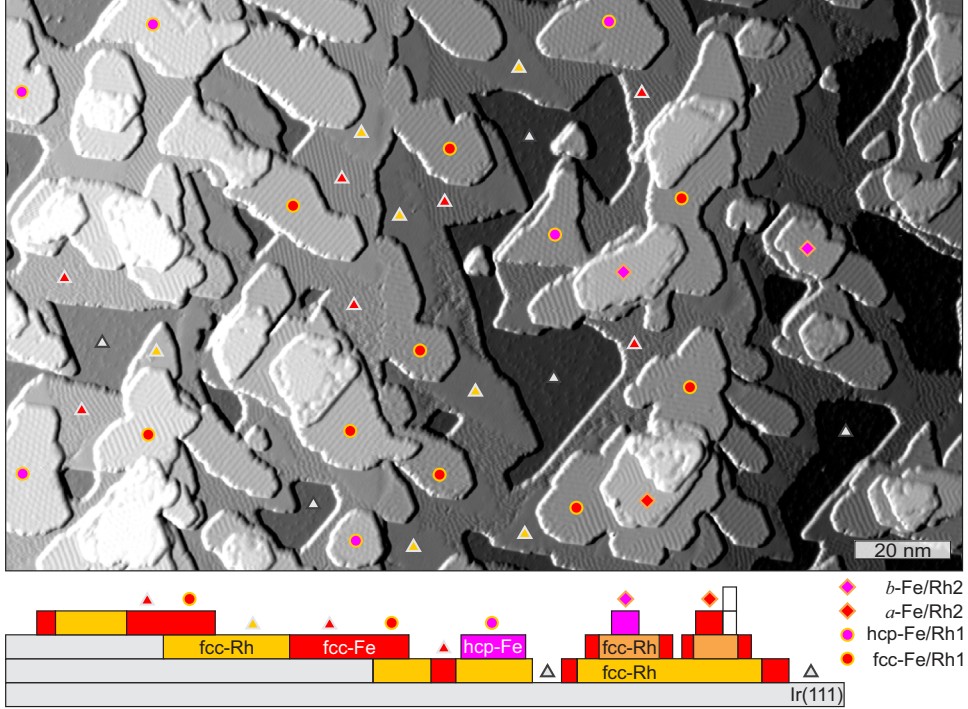

**Fig. 1 | Fe/Rh films on Ir(111).** Overview SP-STM image (partially differentiated constant-current data) of about 0.3 atomic layers of Fe on about 0.5 atomic layers of Rh on Ir(111). Below is a side-view sketch, colored dots indicate a specific layer

sequence. (Measurement parameters: $U = +50$ mV, $I = 1$ nA, $B = -2$ T, $T = 4.2$ K, Cr-bulk tip).

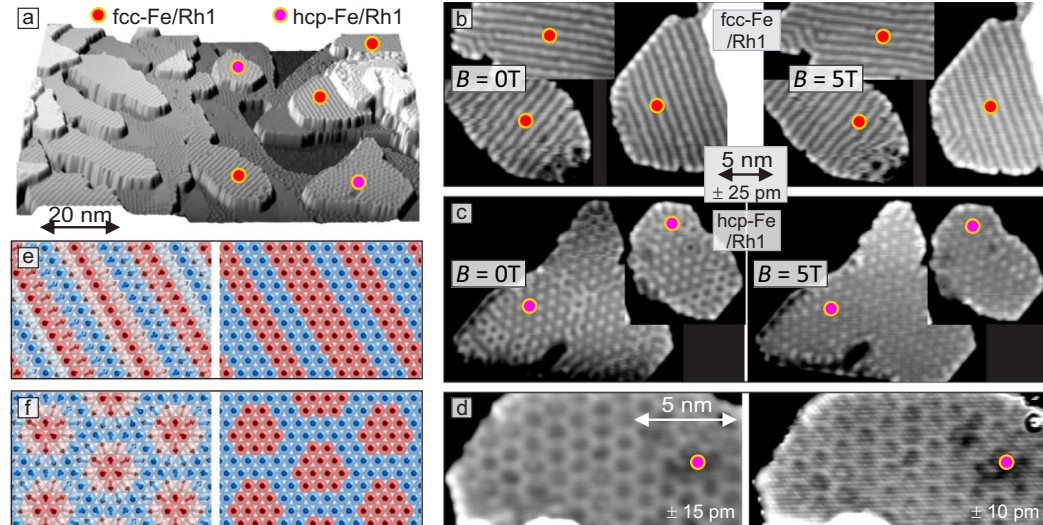

**Fig. 2 | Magnetic states of Fe/Rh1. a** Overview SP-STM image (partially differentiated constant-current data) of Fe/Rh/Ir(111). **b, c** Constant-current data without external magnetic field and with applied magnetic field of $B = 5$ T of fcc- and hcp-stacked Fe-ML/Rh1 islands as indicated in the overview image. **d** Constant-current images of the same hcp-Fe/Rh1 sample area without and with atomic resolution. **e, f** Sketches of possible magnetic states with the experimentally determined unit cells; for details on the construction of these magnetic states see methods section and for SP-STM simulations see Supplementary Figs. 1 and 2 (measurement parameters: **a, b, c**, $U = +50$ mV, $I = 1$ nA; d, $U = +34$ mV, $I = 5.3$ nA, $B = -4$ T; all: $T = 4.2$ K; Cr-bulk tip sensitive to out-of-plane sample magnetization components, as deduced from the observed magnetic contrast and symmetry considerations; the height range of the images is given in pm.).

structures with the experimentally found period of the out-of-plane magnetization components (nearly 5 atomic rows) are shown in Fig. 2e.

The two hcp-Fe/Rh1 islands of Fig. 2c exhibit a hexagonal magnetic superstructure. Areas with bright dots and areas with dark dots coexist in the virgin state (left images). This suggests the presence of two inverted magnetic domains, reminiscent of the hexagonal nanoskyrmion lattice in hcp-Fe/Ir(111)[23]. The application of an external magnetic field of 5 T changes the magnetic pattern within the islands and only the bright dots remain (right image), indicating a switching of one domain in applied magnetic field. Figure 2d shows another hcp-Fe/Rh1 area imaged without and with atomic resolution in an applied magnetic field. The analysis of the constant-current image with atomic resolution demonstrates that the magnetic state is not strictly commensurate, however, a hexagonal superstructure with 27 atoms is a good approximation of the magnetic unit cell. Assuming a quasi-continuous rotation of the magnetization leads to the skyrmion lattice state shown in Fig. 2f (left) with lattice vectors of 1.41 nm perpendicular to the close-packed atomic rows. However, a collinear spin arrangement as sketched in Fig. 2f (right) would also be in accordance with the experimental data. Both displayed magnetic states have a non-vanishing net magnetic moment and two inversional domains are possible, in agreement with the two different patterns observed in the magnetic virgin state and the switching in applied magnetic fields.

To investigate the Fe-MLs on Rh2 in more detail a sample with more Rh was prepared (see overview image presented in Fig. 3a). For the two different stackings of Fe/Rh2 we again observe a uniaxial and a hexagonal magnetic superstructure. We want to first focus on the uniaxial state in the stacking that we name *a*-Fe, see Fig. 3b. Similar to fcc-Fe/Rh1, also in *a*-Fe/Rh2 the uniaxial magnetic state occurs in its three possible rotational domains and in the left image of Fig. 3b, we observe sharp transitions between them. Again the magnetic contrast is the same for all three rotational domains, and based on symmetry considerations we conclude that the out-of-plane sample magnetization components give rise to this pattern. Atomic resolution data reveal that the magnetic state is strictly commensurate with a periodicity of exactly 4-atomic rows, i.e., 0.94 nm. This suggest the presence of a 90° spin spiral or an up-up-down-down (*uudd*) state in this

*a*-Fe/Rh2, see Fig. 3d. The right image of Fig. 3b shows a closer view of this magnetic state imaged at a bias voltage of only a few mV. In this image the observed stripes have half the distance compared to those in the left image, i.e., two atomic rows. These different appearances of a magnetic state in (SP)-STM is characteristic for the *uudd* state, as confirmed by the spin-resolved vacuum density of states calculations for this system shown as insets to Fig. 3b (see Supplementary Fig. 3 for further information). This collinear *uudd* state is also the ground state of an Fe-ML on a Rh(111) single crystal surface[28] and of an Fe-ML sandwiched between a Rh overlayer and Ir(111), with small deviations from the collinear state due to strong DMI[29].

Figure 3c shows two constant-current images of a *b*-Fe/Rh2 island with a hexagonal magnetic superstructure. Between the images the adsorbed cluster near the center of the island was moved with the tip. We find that the details of the magnetic texture have changed between the images. Also across the island the magnetic pattern changes, but we find several areas with a honeycomb pattern that appear to be commensurate magnetic domains. Analysis of the symmetry reveals that two different rotations of this superstructure occur (see white lines in the right image). The constant-current image with atomic resolution (see inset) demonstrates that the hexagonal magnetic state has 19 atoms in the unit cell. The lattice vectors are 1.18 nm long, the magnetic unit cell is rotated by the angle of ±36.65° with respect to the atomic lattice, and the non-collinear skyrmion lattice or the corresponding collinear state as sketched in Fig. 3e are possible magnetic structures. Fe monolayers on the rarely observed hcp-Rh monolayer, on Rh/hcp-Rh, and on Rh3 also exhibit nano-scale magnetic order with slight variations (see Supplementary Figs. 4–6).

Measurements performed with Fe-coated W tips, which are typically sensitive to the samples' in-plane magnetization components in zero field, suggest that the experimentally observed spin textures in Fe-MLs in contact with Rh are dominantly out-of-plane (see Supplementary Fig. 7). A quantification of the experimentally observed magnetic signals is not straightforward, because in addition to the tunnel magnetoresistance (TMR), which is the basis for SP-STM, also other effects can contribute to the tunnel current, such as tunnel anisotropic MR (TAMR)[30], non-collinear MR (NCMR)[31], or effects stemming from the position dependent magnetic polarization of the

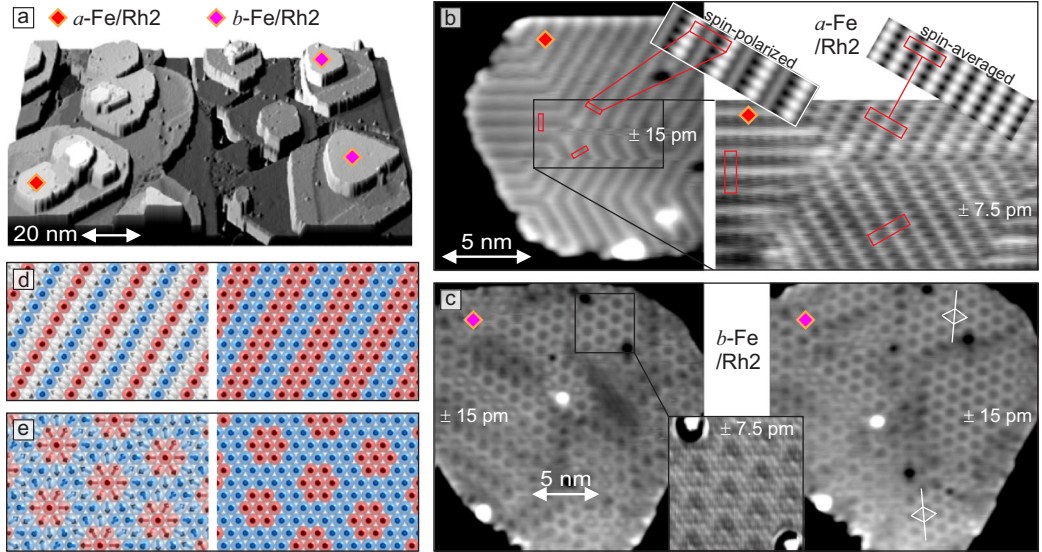

**Fig. 3 | Magnetic states of Fe/Rh2. a** Overview SP-STM image (partially differentiated constant-current data) of Fe/Rh/Ir(111). **b** Constant-current images of the *a*-Fe/Rh2 island indicated in the overview image (rotated with respect to the overview image; note that near the middle of the island a rotational domain has switched direction during imaging (left image, horizontal scan lines)); the two insets show calculated SP-STM images based on the local vacuum density of states of the *uudd* state with different spin-polarization of the tip, see Supplementary Fig. 3 for details; while for the two STM images we cannot derive the size of the respective spin-polarization of the tunnel current, the comparison with the calculated STM images suggests that it is negligible for the right measurement. **c** Constant-current images of a *b*-Fe/Rh2 island before and after moving the central

adsorbed cluster (this is the island in the bottom right of the overview image, rotated with respect to the overview image). **d, e** Sketches of possible magnetic states with the experimentally determined unit cells; for details on the construction of these magnetic states see methods section and for SP-STM simulations see Supplementary Figs. 1,2. (Measurement parameters: a $U = +15$ mV, $I = 3.3$ nA; b, left $U = +41$ mV, $I = 2.8$ nA, right $U = +4$ mV, $I = 5.1$ nA; c, left $U = +10$ mV, $I = 3.3$ nA, right $U = +11$ mV, $I = 2.0$ nA, inset $U = +34$ mV, $I = 5.3$ nA; all: $B = -4$ T, $T = 4.2$ K; Cr-bulk tip sensitive to out-of-plane sample magnetization components, as deduced from the observed magnetic contrast and symmetry considerations; the height range of the images is given in pm.).

Rh atoms, as in the *uudd* state[29,32] (see Supplementary Fig. 3). As a consequence we cannot accurately determine the details of the spin arrangement within a respective magnetic unit cell, e.g., the precise angles between nearest-neighbor moment pairs. However, our experiments have revealed the size and the symmetry of the different magnetic states down to the atomic scale.

## First-principles calculations

To understand the experimentally observed magnetic ground states and the responsible magnetic interactions we have performed first-principles electronic structure calculations based on density functional theory (DFT). Spin spirals are the general solution of the classical Heisenberg model on a periodic lattice. In order to scan a large part of the magnetic phase space and to obtain the exchange constants we calculate by DFT the energy dispersion $E(\mathbf{q})$ of flat spin spirals for all four experimentally studied systems, i.e., the fcc- and hcp-stacked Fe monolayers on Rh mono- and double-layers in fcc stacking (Rh1 and Rh2) on Ir(111) (see methods for computational details and Supplementary Table 1 for relaxed interlayer distances).

The energy dispersion of all four systems looks similar (Fig. 4 and Supplementary Fig. 8): the ferromagnetic state ($\bar{\Gamma}$ point) is a local energy maximum and there are energy minima for spin spirals along both high-symmetry directions with periods of about $\lambda = 1.9 - 1.1$ nm ($q = |\mathbf{q}| \approx 0.14 - 0.25 \times 2\pi/a$). The row-wise antiferromagnetic state ($\bar{M}$ point) and the Néel state ($\bar{K}$ point) are much higher in energy. This energy dispersion originates from a small ferromagnetic nearest-neighbor Heisenberg exchange and strong frustration with antiferromagnetic interactions for second- and third-nearest neighbors (see Supplementary Table 2 for values). We find that the exchange frustration is considerably stronger in the Fe/Rh2 systems, and also in the fcc-Fe systems compared to hcp-Fe stacking. In the spin spiral dispersions this is reflected by deeper spin spiral energy minima and a shift to larger values of **q**, i.e., shorter spin spiral periods.

The experimental observation of the *uudd* state as well as two-dimensionally modulated magnetic states indicates that higher-order interactions play a role for the ground state formation in our system. To obtain the values for the higher-order interactions for our films we calculate the energy of several prototypical multi-Q states, i.e., two collinear *uudd* states (Fig. 4d, e) and the triple-Q state (Fig. 4f)[18,20]. A comparison to their respective 1Q states yields the value of the three different four-spin interactions (see methods). We find exceptionally large higher-order exchange constants of up to nearly 5 meV (Supplementary Table 3), confirming that higher-order exchange interactions (HOI) beyond Heisenberg exchange play an important role in this system.

The hexagonal magnetic states found in the experiments with unit cells on the order of a nanometer (cf. Figs. 2f and 3e) have motivated us to construct hexagonal skyrmion lattices (SkX). We use a normalized superposition of three symmetry-equivalent cycloidal spin spirals, i.e., their *Q*-vectors have equal length, 120° with respect to each other, and normalized amplitudes (see methods for details). In contrast to the *uudd* and the triple-Q state, these larger sized SkX are not expected to have exactly the same exchange energy as their constituting 1Q states in the absence of higher-order interactions. In particular, we calculate a SkX with the hexagonal 27-atom unit cell (27-SkX, see Fig. 4g) observed experimentally for hcp-Fe/Rh1, which is obtained by choosing the spin spiral vectors along the three equivalent $\bar{\Gamma}\bar{K}$ directions of the 2D BZ with a period of 4.5 nearest-neighbor distances, i.e., $q \approx 0.22 \times 2\pi/a$. We find that for both stackings of Fe/Rh1 this 3Q state has a significantly lower energy than the corresponding 1Q state; the same is true for the 19-SkX in the Fe/Rh2 systems (see Supplementary Fig. 8).

Up to now, we have considered uniaxial spin spirals and their superposition states, most of which have a non-collinear spin texture. To address also the possibility of fully collinear magnetic order we investigate uniaxial and hexagonal collinear states that we derive from their non-collinear counterparts by projecting the magnetic moments

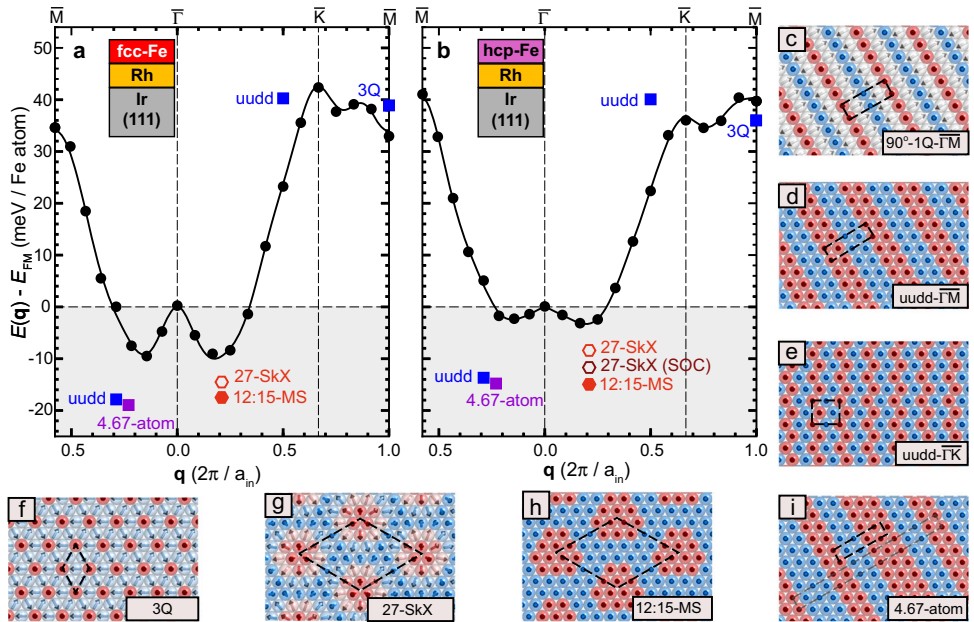

**Fig. 4 | DFT total energies for various spin structures in Fe/Rh/Ir(111).**
**a, b** Energy dispersion $E(\mathbf{q})$ of flat cycloidal spin spirals for fcc-Fe/Rh/Ir(111) and hcp-Fe/Rh/Ir(111), respectively, calculated by means of DFT along the two high-symmetry directions of the two-dimensional Brillouin zone. The black circles denote DFT total energies including spin–orbit coupling (SOC), i.e., the DMI and

MAE. Black lines represent a fit to the Heisenberg model including the contributions of DMI and MAE. The DFT total energies of the spin structures given in **d–i** are shown by symbols at the $q$-values of the respective 1Q states as indicated in the figure. **c–i** Sketches of the considered spin structures, their magnetic unit cells are indicated.

onto the $z$-axis perpendicular to the film (see methods for details). In this way not only the 90° spin spirals turn into *uudd* states, but also arbitrary spin spirals result in collinear states such as the one with a 4.67-atom period, which is displayed in Fig. 4i. The SkX states transform to mosaic states (MS) in the same way, i.e., from the 27-SkX we obtain the 12:15-MS state with 12 moments pointing in one direction and 15 pointing in the opposite direction (Fig. 4g, h). For both stackings of Fe/Rh1 the *uudd*, the uniaxial 4.67-atom state, and the hexagonal 12:15-MS have a significantly lower energy compared to their non-collinear counterparts (for Fe/Rh2 the collinear 7:12-MS also has a lower energy than the corresponding non-collinear 19-SkX, see Supplementary Fig. 8). Note, that we only constrain the direction of the magnetic moments in our DFT calculations while the magnitude is self-consistently determined. However, magnetic states with a significantly modulated moment are highly unfavorable due to the Stoner exchange and the large Fe moments of about 2.8 μ$_B$.

We find that including spin–orbit coupling (SOC), i.e., allowing for the DMI and the magnetocrystalline anisotropy energy (MAE), leads to only small changes of the energies of the magnetic states, e.g., the 27-SkX is lowered by about 3.3 meV (Fig. 4b). This is expected because Rh is a 4$d$ element with a moderate spin–orbit coupling strength (see Supplementary Figs. 9–11 and Supplementary Table 4 for values). Note, that the symmetry of the interfacial DMI favors Néel-type SkX, while Bloch-type SkX do not gain energy. To test the degree of collinearity, we have performed a DFT total energy calculation for a continuous rotation of the magnetic moments from the 12:15-MS state to the 27-SkX state (Fig. 5). If we neglect SOC, the perfectly collinear 12:15-MS state is energetically lowest in our calculation. Upon including SOC, the energy difference to the 27-SkX decreases and we obtain a small energy minimum in the vicinity of the 12:15-MS state. This must be due to the DMI that prefers canted states and apparently overcompensates the MAE that favors collinear order. However, the canting corresponds to only 4° to 11° of the magnetic moments from the surface normal, i.e., a very small degree of non-collinearity (for a similar calculation regarding the *uudd* state see Supplementary Fig. 12, which shows that the canting angle is below 2° in that case).

We have simulated SP-STM images for the proposed spin structures based on the model of ref. 33. For an out-of-plane magnetized tip, see Fig. 5b–d, all of them exhibit a hexagonal pattern. However, the collinear or slightly canted 12:15-MS states exhibit triangular shapes and thus break the six-fold symmetry, similar to the experimentally observed magnetic superstructures (for other tip magnetization directions and other magnetic states see Supplementary Figs. 1 and 2).

**Atomistic spin model**
In order to gain deeper insight into the underlying magnetic interactions, which determine the total energies of different spin structures found by DFT we have studied an extended Heisenberg model with all interaction constants determined from DFT (see methods). Figure 6 shows the energy of selected magnetic states for fcc- and hcp-Fe/Rh1 calculated based on the atomistic spin model (see Supplementary Fig. 13 for Fe/Rh2). The spin structures have been chosen such that a direct comparison of non-collinear (spin spiral or SkX, see top axis) and corresponding collinear (*uudd* or MS, see bottom axis) states is possible. The colored blocks indicate spin textures constructed from the same $Q$-values. For exact multi-Q states such as the *uudd* state along $\overline{\Gamma M}$ the energy due to the pair-wise exchange interaction is equal to that of the corresponding spin spiral state (90°-1Q-$\overline{\Gamma M}$) and any total energy difference arises due to HOI (see e.g., first two states in Fig. 6). Also other states with the same set of $Q$-vectors, such as the pair of non-collinear 77°-1Q-$\overline{\Gamma M}$ and collinear 4.67-atom state, or the 40°-1Q-$\overline{\Gamma K}$, the 27-SkX, and the 12:15-MS state, are nearly degenerate with respect to the exchange term (Fig. 6); however, small differences arise because they are not true superposition states of symmetry-equivalent spin spirals but also contain higher harmonic components (see methods).

The trend of DFT total energies is captured by the spin model (red and black symbols in upper panels in Fig. 6, respectively). The small quantitative deviations for the hexagonal states might be due to beyond nearest-neighbor HOI not taken into account here. While the exchange contribution to the total energy is large (see bottom panels) its variation between different states is rather small (≈15 meV/Fe atom). Regarding the HOIs, the three-site four-spin

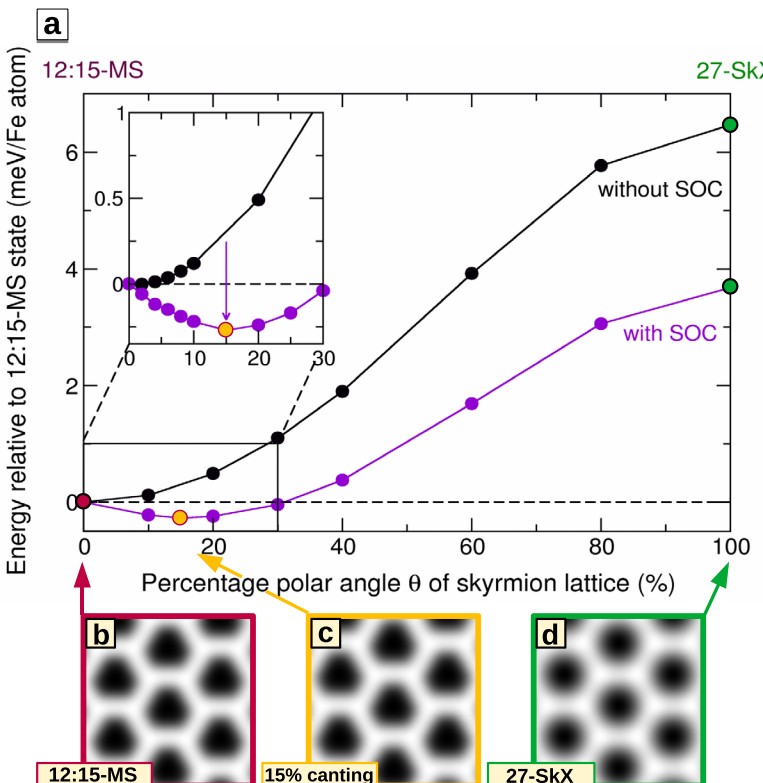

**Fig. 5 | Degree of collinearity of the 12:15-MS state in hcp-Fe/Rh/Ir(111). a** DFT total energies of magnetic states along the geodesic path in spin space from the collinear 12:15-MS state into the non-collinear 27-SkX without (black circles) and with spin–orbit coupling (purple circles). The relative polar angle $\theta$ is defined for every atom as $\theta(x) = \theta_0 + x(\theta_f - \theta_0)$ with $x \in [0, 1]$ where the value $x = 0$ is set for the collinear 12:15-MS state (red filled circle) and $x = 1$ for the non-collinear hexagonal 27-SkX state (green filled circle). $\theta_f$ refers to the final value of every magnetic moment in the 27-SkX, whereas $\theta_0$ is set to 0° for upward pointing moments (180° for downward pointing moments). The in-plane angles $\phi$ of the atoms are not changed with the variation of $\theta$, but kept fixed to the values of the 27-SkX. The solid lines connecting the DFT data points serve as a guide to the eye. **b**–**d** Simulated SP-STM images of the 12:15-MS state, the energetically lowest 15% canted 12:15-MS state, and the 27-SkX with a tip magnetization in $-z$ direction and 50% spin-polarization based on the model of ref. 33.

interaction has the largest contribution and it leads to large energy differences between the states ($\approx$25 meV/Fe atom). One can directly see that it favors the collinear over the corresponding non-collinear states (filled and open data points, respectively). The energy due to biquadratic and four-site four-spin interaction displays a variation on an energy scale of about 10 meV/Fe atom. Their contributions nearly add up to zero energy difference between different states. Therefore, the three-site four-spin interaction dominates the trend in total energy and the interplay of the exchange interaction and the three-site four-spin interaction is decisive for the magnetic ground state of the Fe films (similar conclusions can be drawn for Fe/Rh2, cf. Supplementary Fig. 13).

As the occurrence of multi-Q states that are collinear has not been discussed before we analyze under which conditions they can become the ground states. We have calculated the energy of SkX and MS states as a function of the length of the constituting $Q$-vectors based on the atomistic spin model. The black lines in Fig. 7 indicate the total energy of the respective states with their $Q$-vectors running along the two indicated high-symmetry directions, just as in the spin spiral dispersions shown in Fig. 4. The shape of the non-collinear 3Q energy dispersion is similar to that of the 1Q spin spirals. The contributions from the exchange energy (purple) and the three-site four-spin interaction (green) are roughly mirrored at $E = 0$ when the DFT parameters for hcp-Fe/Rh2 are used; the energies of the other investigated systems are very similar because their individual energy contributions do not deviate significantly (see Supplementary Fig. 14 for hcp-Fe/Rh1). However, in contrast to the parabolic shape of the exchange and 3-spin energies of SkX states

near $\bar{\Gamma}$, the energy contributions of the MS states change linearly around the center of the Brillouin zone. For hcp-Fe/Rh2, as used in Fig. 7, this leads to a preference of the collinear MS state for $q < 0.5 \times 2\pi/a$. Only for longer $Q$-vectors non-collinear SkX states are preferred. The minima of the total energy for the two high-symmetry directions represent collinear MS states, driven by the different energy dependencies of the frustrated exchange and the three-site four-spin interaction for non-collinear and collinear states (a similar comparison is shown for spin spiral states and collinear uniaxial states in Supplementary Fig. 15). The experimentally found hexagonal magnetic states are close to the energy minimum along $\bar{\Gamma}\bar{K}$, in line with the results from the DFT that the magnetic states are collinear.

## Discussion

The Fe-ML in contact with Rh exhibits a variety of complex zero-field nano-scale spin structures, depending on the number of Rh layers and the stacking sequence. They originate from the subtle interplay of frustrated exchange interactions and higher-order terms. In particular the recently proposed three-site four-spin interaction[12,28] is essential in these systems and cannot only stabilize uniaxial states such as the *uudd* state[28] but also spontaneous nano-scale hexagonal spin states. The DMI promotes non-collinear spin structures, however, for Fe/Rh interfaces it plays a minor role since the spin–orbit coupling strength of the 4$d$ transition-metal Rh is rather small. In contrast, the large positive three-site four-spin interaction favors collinear states in the Fe-ML. Therefore, both the uniaxial magnetic states in fcc-stacked Fe, and the hexagonal spin structures discovered for hcp-stacked Fe will exhibit at most a

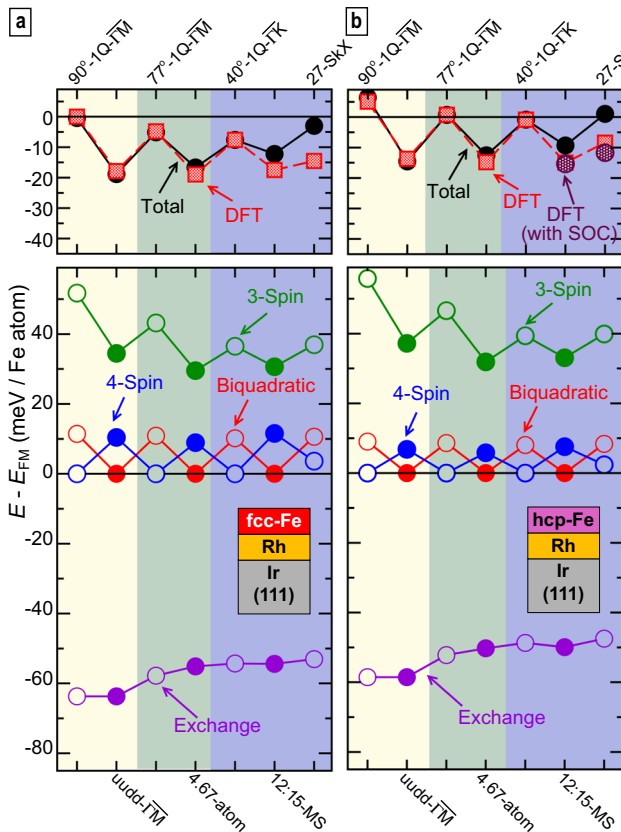

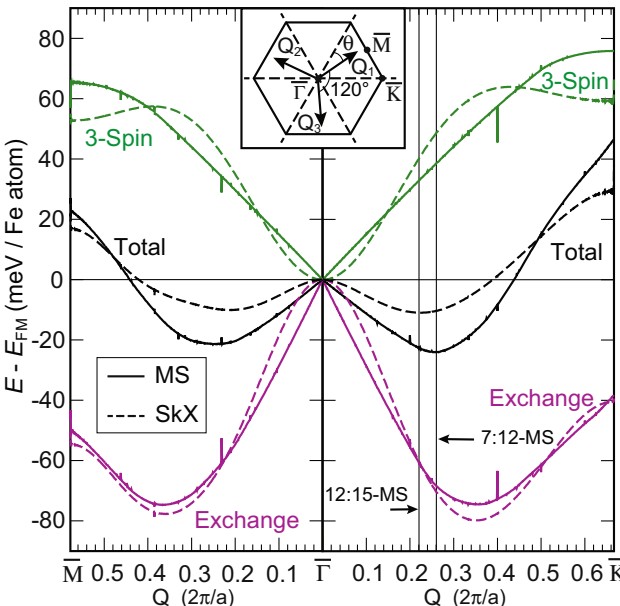

**Fig. 6 | Comparison of selected magnetic states. a, b** Upper panels show the total energies for fcc-Fe/Rh/Ir(111) and hcp-Fe/Rh/Ir(111), respectively, with respect to the FM reference state; red squares are DFT values (violet dots include spin–orbit coupling), black circles are obtained via the atomistic spin model with DFT parameters for the magnetic interactions. On the upper axis the non-collinear states and on the lower axis the corresponding collinear states are specified (background colors serve to group the states, which have the same $Q$-vectors). In the lower panels the total energy of the spin model (black circles in upper panel) is decomposed into the contributions from the Heisenberg exchange, the two-site four-spin interaction (biquadratic), the three-site four-spin interaction (3-Spin) and the four-site four-spin interaction (4-Spin). Filled (open) circles indicate collinear (non-collinear) states. Note that for the calculation of total energies within the spin model the DMI and the MAE were taken into account (not shown here). The lines connecting the data points serve as a guide to the eye.

**Fig. 7 | Energy contributions to SkX and MS states.** Plot of the energy contributions from the exchange and three-site four-spin interaction to the total energy $E(Q)$ for mosaic (MS) and skyrmion lattice (SkX) states for **Q** along the $\overline{M\Gamma K}$ direction. Energies were obtained in the spin model with DFT parameters of hcp-Fe/Rh2/Ir(111). Note, that the energy contributions from the biquadratic, the four-site four-spin interaction, the DMI, and the MAE are not displayed but included in the total energy. The spikes in the energy curves for the MS states originate from changes of the local spin structure on the discrete atomic lattice due to taking only the $z$-component of the magnetic moments in its construction (see methods). Inset: Sketch of the 2D Brillouin zone with the three **Q**-vectors: $Q_1$, $Q_2$, and $Q_3$ used to obtain the SkX along the $\overline{\Gamma M}$ direction; to obtain the SkX for the $\overline{\Gamma K}$ direction the $Q$-vectors are rotated by an angle $\theta = 30°$ (see methods for details).

small non-collinearity with canting angles of a few degrees, as inferred from both DFT and atomistic spin model calculations.

The hexagonal magnetic states discovered here do not resemble the commonly expected non-collinear skyrmion lattices but can be characterized as two-dimensionally modulated collinear multi-Q states, a new class of magnetic order. Owing to their nano-scale size we anticipate that these novel magnetic states exhibit interesting transport properties to be revealed by future work. Not only periodic nano-scale magnetic textures, but also local magnetic perturbations in otherwise collinear states can be governed by such higher-order terms[34,35]. In particular in the growing field of spintronics using antiferro-[36] and ferrimagnets[37] an impact on transport properties such as the anomalous Hall effect[38,39] or the anisotropic magnetoresistance is expected depending on the detailed spin configuration in domain walls[40] or other topological defects such as skyrmions. Significant higher-order interactions have been reported in a range of materials including transition-metal interfaces[7,28,41], rare-earth compounds[26] as well as two-dimensional van der Waals magnets[35,42–44], which are therefore all potential candidates for the new type of spontaneous magnetic order proposed in our work.

## Methods

### Sample preparation

The experiments were performed in a multi-chamber ultra-high vacuum system. Ir(111) was cleaned by sputter and annealing cycles and occasional annealing in oxygen. Rh and Fe were evaporated from rods by electron beam heating and deposited onto the clean Ir surface within about 20 min of the last annealing, i.e., well above room temperature.

### Assignment of layers and stackings

The fcc-Fe monolayer (ML) directly on Ir(111) can easily be identified by its exceptional square nanoskyrmion lattice magnetic ground state occurring in three rotational domains[7]. The Rh monolayer (ML) dominantly grows in a single stacking and because it connects smoothly to the fcc-Fe-ML/Ir(111) we conclude that it is also in fcc stacking (Rh1). Many elongated Fe islands on Rh1 are observed at buried step edges, and it appears as if their growth is induced by the adjacent fcc-stacked ML on the upper terrace, which is why we conclude they are fcc-Fe/Rh1 islands. hcp-Fe/Rh1 is also present, and it is typically found on free-standing Rh1 islands.

Some exposed Fe patches are in total three atomic layers high relative to the Ir(111) surface and almost all of them are still pseudomorphic. The lattice constants of Rh and Ir are very similar, thus pseudomorphic growth is likely also for thicker Rh layers. In contrast, Fe is subject to large strain and we expect that an Fe-double layer (DL) would form dislocation lines, similar to the Fe-DL directly on Ir(111)[45]. These considerations lead to the conclusion that the areas with a thickness of three atomic layers consist of an Fe-ML on a Rh-DL. In the vicinity of these three ML high islands we always find a rim of fcc-Fe/Rh1 and therefore we assume that the Rh-DL consists of fcc-Rh/Rh1 (Rh2).

Owing to their distinct magnetic states the different Fe-stackings can be easily identified in all experiments by their magnetic state.

## Construction of SkX and MS states

The normalized magnetic moment $\mathbf{m}_i^\alpha$ at lattice site $\mathbf{R}_i$ is given for a single spin spiral with a spiral vector $\mathbf{Q}_\alpha$ by

$$\mathbf{m}_i^\alpha = \left( \mathbf{e}_z \cos(\mathbf{Q}_\alpha \cdot \mathbf{R}_i) - \frac{\mathbf{Q}_\alpha}{|\mathbf{Q}_\alpha|} \sin(\mathbf{Q}_\alpha \cdot \mathbf{R}_i) \right), \qquad (1)$$

where $\mathbf{e}_z$ is the unit vector along the $z$ direction, i.e., perpendicular to the surface. The spin structure of the hexagonal skyrmion lattice (SkX) is given by the normalized superposition of the three spin spirals with $\mathbf{Q}_1$, $\mathbf{Q}_2$, and $\mathbf{Q}_3$, which exhibit an angle of 120° with respect to each other (cf. inset of Fig. 7):

$$\mathbf{m}_i^{\text{SkX}} = \frac{\mathbf{m}_i^1 + \mathbf{m}_i^2 + \mathbf{m}_i^3}{|\mathbf{m}_i^1 + \mathbf{m}_i^2 + \mathbf{m}_i^3|}. \qquad (2)$$

Note, that in addition to the length $Q = |\mathbf{Q}_1| = |\mathbf{Q}_2| = |\mathbf{Q}_3|$, we can vary the rotation of the three spin spiral vectors by an angle $\theta$ with respect to the high-symmetry directions of the 2D BZ (cf. inset of Fig. 7). In our DFT calculations we constrain the directions of the magnetic moments according to the construction of the SkX, but their magnitude is obtained self-consistently.

The corresponding mosaic state (MS) is constructed by taking only the normalized $z$-component of the SkX state, i.e.:

$$\mathbf{m}_i^{\text{MS}} = \frac{(\mathbf{m}_i^{\text{SkX}} \cdot \mathbf{e}_z)}{|\mathbf{m}_i^{\text{SkX}} \cdot \mathbf{e}_z|} \mathbf{e}_z \qquad (3)$$

and has only magnetic moments pointing up or down along the out-of-plane direction of the surface.

In an analogous way, we construct a collinear counterpart to a cycloidal spin spiral (1Q) state, $\mathbf{m}_i^{1Q}$:

$$\mathbf{m}_i^{\text{col}} = \frac{(\mathbf{m}_i^{1Q} \cdot \mathbf{e}_z)}{|\mathbf{m}_i^{1Q} \cdot \mathbf{e}_z|} \mathbf{e}_z \qquad (4)$$

## First-principles calculations

In this work we used the full-potential linearized augmented plane wave (FLAPW) method as implemented in the FLEUR code (https://www.flapw.de/) to calculate both the energy dispersion of spin spirals and the two *uudd* states and the 3Q state for Fe/n-Rh/Ir(111) film systems. A spin spiral state is characterized by a single wave vector $\mathbf{q}$ from the two-dimensional Brillouin zone (2D BZ) and the magnetic moment of an atom at lattice site $\mathbf{R}_i$ is given by $\mathbf{M}_i = M(\cos(\mathbf{q} \cdot \mathbf{R}_i), \sin(\mathbf{q} \cdot \mathbf{R}_i), 0)$. In all calculations we used the in-plane lattice constant of Ir obtained within the local density approximation (LDA), i.e., 2.70 Å[46]. The inclusion of exchange-correlation effects was also done by means of the LDA potential with the interpolation developed by Vosko, Wilk and Nusair (VWN)[47]. Structural relaxations for the four systems were performed in the ferromagnetic state by using a symmetric film consisting of nine Ir layers and one Fe/Rh bilayer (Fe/Rh/Rh trilayer) on each side of the film. Here, the cutoff for the basis functions was set to $k_{\max} = 4.1$ a.u.$^{-1}$, 90 $k$-points in the irreducible part of the two-dimensional BZ were used and a mixed LDA-GGA exchange-correlation potential as described in ref. [48] was applied. The muffin tin radii were chosen as 2.31 a.u. for Rh and Ir and a slightly smaller value of 2.23 a.u. was taken for Fe. The relaxed interlayer distances for the four Fe/n-Rh/Ir(111) systems are given in Supplementary Table 1.

In FLEUR the generalized Bloch theorem is applied to perform self-consistent calculations of the energy dispersion of spin spirals in the chemical unit cell within the scalar-relativistic approximation[49]. For

Fe/n-Rh/Ir(111) we used 1936 $k$-points in the full two-dimensional BZ and a large basis set cutoff $k_{\max} = 4.1$ a.u.$^{-1}$. The energy contribution due to DMI for every spin spiral state was calculated based on the self-consistent results including spin–orbit coupling (SOC) within first order perturbation theory[50].

For spin spiral calculations, we have used asymmetric films with 4 (5) Ir layers for Fe/Rh/Rh/Ir(111) (Fe/Rh/Ir(111)). We have checked the effect of using 9 Ir layers on the energy dispersion without spin–orbit coupling (Supplementary Fig. 9) and on the DMI contribution obtained by including spin–orbit coupling (Supplementary Fig. 10). Qualitatively, the energy dispersion without spin–orbit coupling does not change upon increasing the substrate thickness, however, the depth of the spin spiral minima is reduced. The DMI is even quantitatively in good agreement for the two substrate thicknesses. The exchange and DMI constants are also very similar (Supplementary Tables 2 and 3).

The MAE for the ferromagnetic state was obtained by including SOC in our calculations[51] using the force theorem for asymmetric Fe/Rh/Ir(111) films with 15 Ir substrate layers and self-consistently for asymmetric Fe/Rh/Rh/Ir(111) films with 9 Ir substrate layers. Since this quantity is usually very small, the basis set cutoff $k_{\max}$ was increased to 4.3 a.u.$^{-1}$ and for the $k$-point mesh we used 1936 $k$-points in the full two-dimensional BZ.

For the evaluation of the HOI parameters (see below) it is essential to use the same number of substrate layers for both single-Q and the multi-Q states, i.e., the two *uudd* states (Fig. 4d, e) and the 3Q state (Fig. 4f). Since the total energies of the latter ones need to be computed within a four-atomic unit cell per layer, one has to choose the film thickness accordingly in order to make the calculations computationally feasible when using an all-electron method. Hence as mentioned before, we used for the calculation of the total energy differences between spin spiral and the corresponding multi-Q states asymmetric films with 4 and 5 Ir layers for Fe/Rh/Rh/Ir(111) and Fe/Rh/Ir(111), respectively. The number of $k$-points for the *uudd* state along $\overline{\Gamma M}$ direction amounts to 168 in the irreducible part of the BZ, to 336 for the *uudd* state along $\overline{\Gamma K}$ direction and to 242 for the non-collinear 3Q state. In the Supplementary Information we also show for the example of the Fe/Rh/Ir(111) system that this approach does not alter the nearest-neighbor (NN) HOI terms much as compared to a calculation with nine Ir layers (Supplementary Table 4).

In order to handle collinear and non-collinear spin structures in large supercells containing up to 200 atoms, we resorted to the projected augmented wave method[52] as implemented in the VASP code (https://www.vasp.at/)[53,54]. The structural parameters have been chosen consistently with those of the FLEUR calculations and the local density approximation[47] was also used for the exchange and correlation part of the potential. The energy cutoff was set to 300 eV for all calculations. The total energy of the 4.67-atom state along $\overline{\Gamma M}$ direction was calculated within its 14-atom commensurate magnetic unit cell including three magnetic periods on a $7 \times 22 \times 1$ Monkhorst-Pack (MP) $k$-point mesh. The two-dimensional BZ of the hexagonal 12:15-MS state and of the corresponding 27-atomic hexagonal SkX (with the complete unit cell containing 189 atoms) was sampled by $5 \times 15 \times 1$ $k$-points. The same $k$-mesh was applied to calculate the energies of spin states for the continuous transformation of the 12:15-MS into the 27-SkX state. For the hexagonal 7:12-MS state as well as the corresponding 19-atomic hexagonal SkX we used $11 \times 11 \times 1$ $k$-points. The density of the $k$-meshes for these states was chosen in such a way that it corresponds to roughly 1/27 and 1/19 of the spin spiral calculations in accordance with the number of Fe atoms within the surface unit cell. The total energies for spin states along the transformation path of the collinear *uudd* state into the 90° spin spiral were calculated on a $14 \times 44 \times 1$ MP $k$-point mesh within the 4-atomic unit cell per layer of the *uudd* state (Supplementary Fig. 12).

The total energies of the skyrmion lattices as well as the energies of the magnetic states along the geodesic path from the collinear 12:15-MS into the corresponding 27-SkX were determined self-consistently

by using the constrained local moment approach with fixed direction of every magnetic moment in the unit cell and only allowing their magnitudes to relax. Starting from these results, spin–orbit coupling effects were added within a subsequent non-self-consistent calculation in which the converged charge density was kept constant. A modulation of the magnetic moment of Fe is energetically very unfavorable due to Stoner exchange. Accordingly the magnitude of the magnetic moment $M$ is found in our DFT calculations to be about 2.8 $\mu_B$ per Fe atom and varies by less than 5% for spin spirals as a function of $\mathbf{q}$ and for all other considered magnetic states such as SkX and MS. The two last types of states exhibit a non-vanishing magnetic moment per unit cell. It amounts to about 14 $\mu_B$ per unit cell for the 27-SkX state (about 8.7 $\mu_B$ for the 12:15-MS) and about 9.8 $\mu_B$ per unit cell for the 19-SkX state (about 14.7 $\mu_B$ for the 7:12-MS).

## Atomistic spin model

The atomistic spin model is given by

$$
\begin{aligned}
H = & -\sum_{ij} J_{ij}(\mathbf{m}_i \cdot \mathbf{m}_j) - \sum_{ij} \mathbf{D}_{ij}(\mathbf{m}_i \times \mathbf{m}_j) - \sum_i K_u (m_i^z)^2 \\
& - \sum_{<ij>} B_1(\mathbf{m}_i \cdot \mathbf{m}_j)^2 - \sum_{<ijk>} Y_1[(\mathbf{m}_i \cdot \mathbf{m}_j)(\mathbf{m}_j \cdot \mathbf{m}_k) \\
& + (\mathbf{m}_j \cdot \mathbf{m}_i)(\mathbf{m}_i \cdot \mathbf{m}_k) + (\mathbf{m}_i \cdot \mathbf{m}_k)(\mathbf{m}_k \cdot \mathbf{m}_j)] + \\
& - \sum_{<ijkl>} K_1[(\mathbf{m}_i \cdot \mathbf{m}_j)(\mathbf{m}_k \cdot \mathbf{m}_l) \\
& + (\mathbf{m}_i \cdot \mathbf{m}_l)(\mathbf{m}_j \cdot \mathbf{m}_k) - (\mathbf{m}_i \cdot \mathbf{m}_k)(\mathbf{m}_j \cdot \mathbf{m}_l)],
\end{aligned}
\tag{5}
$$

where $\mathbf{m}_i = \mathbf{M}_i/M_i$ is the normalized magnetic moment at lattice site $i$ in the Fe layer, $J_{ij}$ are the Heisenberg exchange constants, $\mathbf{D}_{ij}$ is the DMI vector, and $K_u$ is the uniaxial magnetocrystalline anisotropy constant. The two lower lines describe the higher-order exchange interactions (HOI) obtained in fourth-order perturbation theory from a multi-band Hubbard model[12]. These terms are the biquadratic or two-site four-spin ($B_1$), the three-site four-spin ($Y_1$), and the four-site four-spin ($K_1$) interaction. Since these terms arise in fourth-order perturbation theory we treat them in nearest-neighbor approximation as suggested in ref. 12 and indicated in the summation by < . . > . All interaction constants have been determined from the DFT calculations using the FLEUR code and are given in Supplementary Table 3 and Supplementary Tables 2 and 4.

## Determination of HOI constants

To obtain the values for the higher-order interactions for our films we calculate via the FLEUR code the energy of several prototypical multi-Q states since in DFT all interactions are implicitly included within the exchange-correlation functional. As mentioned above, we use the collinear *uudd* states (Fig. 4d, e), which can be viewed as 2Q states resulting from the superposition of two counterpropagating 90° spin spirals (Fig. 4c). This is possible for spin spirals both along the $\overline{\Gamma M}$ (Fig. 4d) and $\overline{\Gamma K}$ (Fig. 4e) direction, resulting in two different *uudd* states[28,55]. The third multi-Q state is the non-collinear non-coplanar 3Q state (Fig. 4f), which arises due to the superposition of three spin spirals at the $\overline{M}$ points of the 2D BZ[18,20].

The HOI lift the energetic degeneracy between the multi-Q and the corresponding single-Q states and can be obtained from a set of coupled equations[12]:

$$
\Delta E_{\overline{M}} = E_{3Q} - E_{\overline{M},1Q} = \frac{16}{3}(2K_1 + B_1 - Y_1) \tag{6}
$$

$$
\Delta E_{\frac{1}{2}\overline{\Gamma M}} = E_{uudd,\frac{\overline{M}}{2}} - E_{\frac{\overline{M}}{2},1Q} = 4(2K_1 - B_1 - Y_1) \tag{7}
$$

$$
\Delta E_{\frac{3}{4}\overline{\Gamma K}} = E_{uudd,\frac{3\overline{K}}{4}} - E_{\frac{3\overline{K}}{4},1Q} = 4(2K_1 - B_1 + Y_1) \tag{8}
$$

The energy differences $\Delta E_{\overline{M}}$, $E_{uudd,\frac{\overline{M}}{2}}$, and $\Delta E_{\frac{3}{4}\overline{\Gamma K}}$ between the multi-Q and spin spiral states are indicated in Fig. 4a, b and Supplementary Fig. 8a, b, the exact values are given in Supplementary Table 5. We have checked the influence on increasing the number of Ir layers from 5 to 9 in our calculation of the HOI constants for Fe/Rh/Ir(111) and found only a small change of the values (see Supplementary Table 5).

We note that the HOI terms need to be taken into account when extracting the Heisenberg exchange constants from fitting the spin spiral energy dispersion. In particular, the first three exchange constants $J_i$ as obtained from the fit neglecting HOI need to be adjusted in the following way[34] since the respective analytical expressions of the spin spiral dispersion are identical:

$$
J_1' = J_1 - Y_1 \tag{9}
$$

$$
J_2' = J_2 - Y_1 \tag{10}
$$

$$
J_3' = J_3 - \frac{1}{2}B_1 \tag{11}
$$

## Data availability

The data presented in this paper are available from the authors upon reasonable request.

## Code availability

The code for the spin model calculations is available from the authors upon reasonable request.

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

## Acknowledgements

We gratefully acknowledge financial support from the Deutsche Forschungsgemeinschaft (DFG, German Research Foundation) via projects no. 402843438, no. 408119516, no. 414321830, and no. 418425860 and computing time provided by the North-German Supercomputing Alliance (HLRN).

## Author contributions

A.K. and K.v.B. performed the experiments and analyzed the data. M.G., S. Haldar, and H.P. performed the first-principles and spin model calculations. M.A.G. and M.G. performed the SP-STM simulations. M.G.,

S. Haldar, and S. Heinze analyzed the theoretical results. M.G., S. Haldar, S. Heinze, and K.v.B. wrote the manuscript. M.G., A.K., S. Haldar, M.A.G., R.W., S. Heinze, and K.v.B. discussed the results and commented on the manuscript.

## Funding

## Competing interests
The authors declare no competing interests.
