## [Peer Review File · Nature Communications]

REVIEWER COMMENTS

Reviewer #1 (Remarks to the Author):

The manuscript by Gutzeit et al., builds on a broad foundation of spin-polarized STM studies from the Hamburg group using the Ir(111) substrate as a platform for nanomagnetism, including work on Fe/Ir(111), Rh/Fe/Ir(111) and here Fe/Rh/Ir(111). Other work has looked at the evolution of magnetic textures in mono-, bi- and trilayer Fe/Ir(111) which exhibit different stacking and/or strain profiles. The permutations allow for tuning of the net spin orbit coupling and the resultant DMI, which favors noncollinear spin textures, and other magnetic interactions as well as frustration. This has yielded a progressively richer set of magnetic phases which can then inform efforts in other materials systems. This is an important frontier in nanomagnetism, which is driven by broad fundamental interest, as well as the potential for novel applications.

The aim of the current study is to demonstrate how the interplay of higher-order magnetic interactions, frustration and DMI can stabilize either noncollinear or collinear ordering, yet produce similar magnetic stripe or lattice textures with nanoscale dimensions. Through a detailed comparison with theoretical calculations, the authors show that what otherwise would be assigned to single-Q helical or 3-Q skyrmion phases with noncollinear ordering of spins, may instead reflect states with collinear ordering (i.e., 'uudd' or '12:15 MS' states respectively).

Both the experimental and theoretical data are of very high quality, and I appreciate the effort made to directly connect the two by modeling the exact same systems measured in the experiments. There remain some disconnects though, which make it hard to follow the authors' reasoning and conclusions. For example, the schematics in Figs. 2e-f for the non-collinear textures don't match the theoretical conclusions that the collinear uudd and 12:15-MS states are more stable. In fact, the manuscript contradicts itself on line 271 of pg 13. The manuscript should be revised for a more coherent and integrated presentation of the experimental and theoretical results.

I would also like to see more hypothesis testing between the experiment and theory. The end result of the experimental part of the manuscript is inconclusive, but it seems there should be a way to use the experimental data to reduce the subset of magnetic textures considered theoretically. For example, the uniform stripe width seems to rule out the '4.67' collinear state. Also, it is unclear if the modulated contrast in Fig. 3b does indeed match the uudd state in Fig. 3d; at a glance it looks more 'udud'-like. Also, I could not readily find any mention of the tip polarization direction, but it would seem that comparing out-of-plane and in-plane sensitive tips would quickly distinguish between SkX and 12:15 MS states; with an in-plane tip only the SkX state should appear double-lobed. This would more directly prove the theoretical conclusions.

Reviewer #2 (Remarks to the Author):

In the manuscript entitled “Nano-scale collinear multi-Q states driven by higher-order interactions”, the authors reported the experimental discovery of various types of single-Q and triple-Q modulated spin textures for Fe/Ru films on Ir(111) by using the spin-polarized STM measurements. On the basis of the corresponding first-principles calculation and additional theoretical analysis with the effective Hamiltonian, the authors concluded that the collinear triple-Q spin state, rather than the conventional non-collinear triple-Q spin state (i.e. Neel-type skyrmion lattice state), is favored in this system mainly due to the contribution from three-site four-spin interactions.

Recently, the search of novel multiple-Q spin textures and their formation mechanisms has been a hot topic in the field of topological magnetism. As the authors admitted, the present experimental data alone cannot distinguish the collinear/non-collinear nature of the observed triple-Q state. Nevertheless, by considering the good theoretical reproduction of the observed single-Q/triple-Q nature and associated spin modulation period, the proposed collinear triple-Q spin texture and associated three-site four-spin interaction mechanism seem plausible. It may provide a new material design criteria for further exploration of nontrivial particle-like spin textures. In this context, I feel that it can potentially be suitable for publication in Nature Communications, if the following issues are appropriately addressed.

1. In the theoretical analysis, the authors demonstrated that the collinear mosaic state has the lower energy than the other candidates shown in Fig. 4. Nevertheless, it doesn't confirm that the collinear mosaic state is the true ground state (There may be other lowest energy state). For example, the authors didn't consider the skyrmion lattice state with different spin swirling manner, or the amplitude-modulated collinear spin state. Why can the authors exclude these possibilities?

2. Since the effective Hamiltonian and associated parameters are obtained here, I naively expect that some micromagnetic (Monte Carlo) simulation may be able to identify the true ground state. Can the authors perform such an additional simulation?

3. In the experimental part, the authors identified two types of Fe/Rh stacking, namely fcc-Fe/Rh and hcp-Fe/Rh, which were proposed to hosts single-Q and triple-Q orders, respectively. How can the authors distinguish the stacking manner in each island?

Reviewer #3 (Remarks to the Author):

This manuscript reports a combined study between experiment and theory for nano-scale magnetic textures on Fe/Rh atomic layers. The authors discovered several different types of uniaxial and hexagonal textures by SP-STM, and discussed their spin configurations and stabilization mechanism by using the first-principles calculations and the spin model analyses. The main finding is the nano-scale collinear spin textures, in which a three-site four spin interaction plays a role. Although the experimental and theoretical results sound reasonable, I do not recommend the publication in Nature Communication by the following reasons.

i. The motivation of this study is not clear. What were the unsolved problems in the previous studies mentioned in the introduction?

ii. Similar nano-scale collinear spin states have been discussed in the literature, e.g., Ref. 28 and 29. What is a new aspect in the spin textures found in this study?

iii. The authors called the collinear multi-Q states a new class of magnetic order. But it looks similar to the so-called magnetic bubbles. Is the difference only in the size?

iv. It is not clear whether the nano-scale collinear states exhibit interesting transport properties. It is well known that noncollinear states like skyrmions lead to intriguing transport, such as the topological Hall effect, by the Berry phase mechanism. In the collinear case, the Berry phase does not play a role. What kind of interesting transport phenomena do the authors expect for the collinear states?

v. The emphasis is on the three-site four spin interaction, but it appears that the biquadratic interaction with $B_1 > 0$ also plays an important role in stabilizing the collinear states. In some previous studies, the biquadratic interaction with $B_1 < 0$ was argued for noncollinear spin textures. The authors are asked to comment on the importance of the positive biquadratic interaction and the reason for the different sign of B_1 .

Response to Reviewer #1:

The manuscript by Gutzeit et al., builds on a broad foundation of spin-polarized STM studies from the Hamburg group using the Ir(111) substrate as a platform for nanomagnetism, including work on Fe/Ir(111), Rh/Fe/Ir(111) and here Fe/Rh/Ir(111). Other work has looked at the evolution of magnetic textures in mono-, bi- and trilayer Fe/Ir(111) which exhibit different stacking and/or strain profiles. The permutations allow for tuning of the net spin orbit coupling and the resultant DMI, which favors noncollinear spin textures, and other magnetic interactions as well as frustration. This has yielded a progressively richer set of magnetic phases which can then inform efforts in other materials systems. This is an important frontier in nanomagnetism, which is driven by broad fundamental interest, as well as the potential for novel applications.

The aim of the current study is to demonstrate how the interplay of higher-order magnetic interactions, frustration and DMI can stabilize either noncollinear or collinear ordering, yet produce similar magnetic stripe or lattice textures with nanoscale dimensions. Through a detailed comparison with theoretical calculations, the authors show that what otherwise would be assigned to single-Q helical or 3-Q skyrmion phases with noncollinear ordering of spins, may instead reflect states with collinear ordering (i.e., 'uudd' or '12:15 MS' states respectively).

Both the experimental and theoretical data are of very high quality, and I appreciate the effort made to directly connect the two by modeling the exact same systems measured in the experiments. There remain some disconnects though, which make it hard to follow the authors' reasoning and conclusions. For example, the schematics in Figs. 2e-f for the non-collinear textures don't match the theoretical conclusions that the collinear uudd and 12:15-MS states are more stable. In fact, the manuscript contradicts itself on line 271 of pg 13. The manuscript should be revised for a more coherent and integrated presentation of the experimental and theoretical results.

We thank the reviewer for the careful evaluation of our work and the very positive assessment of the experimental and theoretical aspects.

Based on the above suggestions we have included sketches of both the non-collinear and the collinear spin texture in the experimental figures (Fig. 2 and Fig. 3). We appreciate this suggestion as it emphasizes that both spin structures are in agreement with the presented experimental data. In addition, we have performed simulations of spin-polarized STM images for the different proposed spin structures and present them in the new Supplementary Figs. S1, S2 and S3.

I would also like to see more hypothesis testing between the experiment and theory. The end result of the experimental part of the manuscript is inconclusive, but it seems there should be a way to use the experimental data to reduce the subset of magnetic textures considered theoretically. For example, the uniform stripe width seems to rule out the '4.67' collinear state. Also, it is unclear if the modulated contrast in Fig. 3b does indeed match the uudd state in Fig. 3d; at a glance it looks more 'udud'-like. Also, I could not readily find any mention of the tip polarization direction, but it would seem that comparing out-of-plane and in-plane sensitive tips would quickly distinguish between SkX and 12:15 MS

states; with an in-plane tip only the SkX state should appear double-lobed. This would more directly prove the theoretical conclusions.

The reviewer is correct that an experiment with in-plane and out-of-plane sensitive magnetic tips on the same sample area would provide additional information. We have performed such an experiment. However, it is difficult to obtain a high-quality series without tip changes, and in the data that we now present as the new Supplementary Fig. S7 identical sample areas are measured without and with applied field (i.e. with in-plane and out-of-plane sensitive Fe-coated W tip), but small tip changes in between images. This data supports the theoretical conclusion that the spin textures dominantly have an out-of-plane component, even if we would be reluctant to draw this conclusion only based on the experimental data presented.

For the data presented in the Main Text a Cr bulk tip was used, which in general can have an arbitrary tip magnetization direction. Following symmetry considerations we have derived that the magnetic signal in the images is dominantly stemming from out-of-plane sample magnetization components (e.g. for homogeneous spin spirals running in different directions an in-plane tip should show a strong difference in the magnetic contrast amplitude depending on the projection of tip and sample magnetization, whereas an out-of-plane tip always images spirals with the same magnetic contrast amplitude even when they run in different directions). Note that two scenarios are possible: either the tip is fully out-of-plane, or the tip is canted but the sample does not have significant in-plane magnetization components. Because in all our measurements even with different micro-tips it looks like the visible magnetization components are out-of-plane, a collinear out-of-plane state seems likely. We have moved the information about the tip sensitivity from the caption to a more prominent position in the Main Text.

In order to compare the experimental data with the considered spin structures we have performed SP-STM simulations, as displayed in the new Supplementary Figs. S1 and S2. The simulated SP-STM image for the out-of-plane magnetization component of the 27-SkX state shows a perfect hexagonal lattice. In contrast, the corresponding simulated SP-STM image of the 12:15-MS state shows a deviation from the C_6 symmetry, in agreement with the experimental SP-STM images (Fig. 2d), which supports our conclusion that this is the magnetic ground state for hcp-Fe/Rh1.

Regarding the 4.67-atom state the experimental data suggests that the magnetic state cannot be fully described by a single q -vector, regardless whether it is non-collinear or collinear. The faint substructure along the stripe suggest small contributions from at least one other q -vector in a different direction. Accordingly, the SP-STM simulations are not in very good agreement with the experimental data. Because our main focus is on the hexagonal states we do not discuss this in detail further in the manuscript.

Regarding the uudd state we have exchanged the right image of Fig. 3b with another one taken at the same sample position. This new image shows a stripe pattern with half of the magnetic period, which is characteristic of the uudd state imaged with a spin-averaging tip, as evidenced by DFT-based SP-STM calculations, which are now also displayed in Fig. 3. Details are explained in the corresponding text and the new Supplementary Fig. S3.

We hope that with our additional data the Reviewer can recommend publication of our manuscript in Nature Communications.

Response to Reviewer #2:

In the manuscript entitled “Nano-scale collinear multi-Q states driven by higher-order interactions”, the authors reported the experimental discovery of various types of single-Q and triple-Q modulated spin textures for Fe/Ru films on Ir(111) by using the spin-polarized STM measurements. On the basis of the corresponding first-principles calculation and additional theoretical analysis with the effective Hamiltonian, the authors concluded that the collinear triple-Q spin state, rather than the conventional non-collinear triple-Q spin state (i.e. Néel-type skyrmion lattice state), is favored in this system mainly due to the contribution from three-site four-spin interactions.

Recently, the search of novel multiple-Q spin textures and their formation mechanisms has been a hot topic in the field of topological magnetism. As the authors admitted, the present experimental data alone cannot distinguish the collinear/non-collinear nature of the observed triple-Q state. Nevertheless, by considering the good theoretical reproduction of the observed single-Q/triple-Q nature and associated spin modulation period, the proposed collinear triple-Q spin texture and associated three-site four-spin interaction mechanism seem plausible. It may provide a new material design criteria for further exploration of nontrivial particle-like spin textures. In this context, I feel that it can potentially be suitable for publication in Nature Communications, if the following issues are appropriately addressed.

We thank the reviewer for the appreciation of our work and answer the issues raised in a point-by-point fashion below. We have further improved our manuscript based on the reviewer's questions.

1. In the theoretical analysis, the authors demonstrated that the collinear mosaic state has the lower energy than the other candidates shown in Fig. 4. Nevertheless, it doesn't confirm that the collinear mosaic state is the true ground state (There may be other lowest energy state). For example, the authors didn't consider the skyrmion lattice state with different spin swirling manner, or the amplitude-modulated collinear spin state. Why can the authors exclude these possibilities?

Based on the symmetry of the interface-DMI, which is present in our system, we can exclude that a helical rotation of the spin texture can lead to a lower energy. The interface-DMI can lower the energy of cycloidal spin rotations with the preferred rotational sense according to the sign of the material-specific prefactor which depends on the electronic structure and can be obtained from DFT. Therefore, only Néel-type skyrmion lattices have been included in the calculations. We have added this information in the main text.

In the case of Fe, the strong intra-atomic exchange interaction strongly disfavors a modulation of the magnetic moment. Therefore, a spin structure with an amplitude-modulated magnetic moment is energetically very unfavorable. Note, that in our DFT calculations of the spin structures we have only constrained the directions of the magnetic moments, however, their magnitudes are self-consistently calculated. From these DFT calculations, we find that the magnetic moment of Fe is about $2.8 \mu_B$ and varies by less than 5% for different magnetic configurations or for Fe atoms with different magnetization directions within a given collinear or non-collinear spin structure. We have also added this information to the main text.

We believe that these slight revisions in the main text clarify the possible spin structures.

2. Since the effective Hamiltonian and associated parameters are obtained here, I naively expect that some micromagnetic (Monte Carlo) simulation may be able to identify the true ground state. Can the authors perform such an additional simulation?

We agree with the reviewer that, if the atomistic spin model reproduces the total DFT energies of all magnetic states, a Monte-Carlo or spin dynamics simulation should identify the true ground state of the system. However, because the different states are close in energy, the atomistic spin model with the parameters obtained from DFT for the Heisenberg exchange, DMI, magnetocrystalline anisotropy, and higher-order exchange is quantitatively not sufficiently accurate. While it can qualitatively well explain the trend observed between different spin structures (cf. upper panels of Fig. 6 in the revised manuscript) it still slightly favors the uudd state over the 4.67 atom state for fcc-Fe/Rh1 (Fig. 6a) and the uudd state over the 12:15-MS state for hcp-Fe/Rh1 (Fig. 6b). These small quantitative differences could be due to neglecting beyond nearest-neighbor higher-order exchange in our spin model.

Nevertheless, the spin model explains qualitatively the preference of collinear over non-collinear spin states (e.g. the uudd state over the 90° spin spiral for fcc-Fe/Rh1 and the 12:15-MS state over the 27-SkX state for hcp-Fe/Rh1, cf. Fig. 6 and Fig. 7 in the revised manuscript for the general discussion of collinear vs. non-collinear states).

In response to the reviewers' comments, we have also performed further DFT calculations for the total energy as a function of the canting angle if one continuously transforms the 12:15-MS state into the 27-SkX state along the geodesic path (new Fig. 5 in the revised manuscript). These calculations confirm that upon neglecting spin-orbit coupling, i.e. without DMI, a perfectly collinear 12:15-MS state has the lowest energy. Upon including spin-orbit coupling, a very shallow energy minimum forms. The non-collinearity at the minimum amounts to angles of individual spins of 4 to 11° from the surface normal. We believe this additional Figure in the main text emphasizes the surprising finding of collinear magnetic order.

The spin model (see Supplementary Fig. S16 in the revised Supplementary Information) qualitatively agrees with these DFT calculations. In particular, the lowest energy is obtained for the collinear 12:15-MS state if the DMI is neglected and an energy minimum for a small non-collinearity appears upon including DMI. Therefore, the comparison with the spin model confirms our interpretation that the collinear multi-Q state is driven by higher-order exchange.

3. In the experimental part, the authors identified two types of Fe/Rh stacking, namely fcc-Fe/Rh and hcp-Fe/Rh, which were proposed to hosts single-Q and triple-Q orders, respectively. How can the authors distinguish the stacking manner in each island?

In Fig. 1 we show an overview image with all the different stackings and layer sequences. In the methods section we guide through our assignment of the different exposed local areas. We rely on the knowledge about the fcc stacking of the Fe monolayer directly on Ir(111) [S. Heinze et al, Nature Physics 7, 713 (2011)]. From there we can identify the fcc-Rh1 which attaches next to the fcc-Fe without the formation of a gap. The fcc-Fe/Rh1 is identified by the smooth transition of fcc-Fe/Ir(111) on an upper terrace to the Fe/Rh1 on a lower terrace. In this manner we can identify several other stackings as detailed in the

methods section and indicated in the experimental images. In the respective methods section we have added the information that after the assignment and identification of the corresponding magnetic states we can easily derive the stacking and layer sequence from the particular observed unique magnetic state of a given Fe/Rh_n/Ir(111).

Response to Reviewer #3:

This manuscript reports a combined study between experiment and theory for nano-scale magnetic textures on Fe/Rh atomic layers. The authors discovered several different types of uniaxial and hexagonal textures by SP-STM, and discussed their spin configurations and stabilization mechanism by using the first-principles calculations and the spin model analyses. The main finding is the nano-scale collinear spin textures, in which a three-site four spin interaction plays a role. Although the experimental and theoretical results sound reasonable, I do not recommend the publication in Nature Communication by the following reasons.

We thank the reviewer for taking the time to evaluate our work. However, we respectfully disagree with the negative assessment of our work. We hope that our comments and answers to the points raised below can convince also this reviewer that our manuscript meets the high standards of Nature Communications.

i. The motivation of this study is not clear. What were the unsolved problems in the previous studies mentioned in the introduction?

The initial motivation of this study was the question whether a pseudomorphic Fe monolayer (ML) would possess different magnetic properties on a thin layer of Rh on the Ir(111) surface as compared to directly on Rh(111). This question was driven by the fact that novel spin structures were previously discovered for an Fe ML on both Ir(111) and Rh(111). The simple answer to this question is yes.

However, during this explorative research study many more interesting aspects and questions appeared. For instance, we discovered that not all spin structures which would naively be identified as skyrmion lattices are really non-collinear as commonly expected. This triggered our extensive theoretical study using DFT and atomistic spin simulations on the driving forces for collinear vs. non-collinear multi-Q states, with the experimental data as basis for magnetic ground states that actually exist in reality.

Our study reveals that the interplay of frustrated pair-wise and higher-order exchange can stabilize such collinear multi-Q states which were previously not observed. Since the required ingredients of exchange frustration and higher-order exchange also occur in other material classes such as, e.g., 2D van der Waals magnets it is quite likely that this type of collinear multi-Q states will be reported in the future for other systems as well.

ii. Similar nano-scale collinear spin states have been discussed in the literature, e.g., Ref. 28 and 29. What is a new aspect in the spin textures found in this study?

Refs. 28 and 29 show experimental evidence of the uudd state for the first time, in the latter reference this state is slightly canted due to significant DMI. Therefore, the reviewer is correct, that the uudd state has been found experimentally before and discussed in detail also from the theoretical side. Also one square and one hexagonal nanoskyrmion lattice has been identified experimentally in the different stackings of Fe/Ir(111).

However, in the present manuscript we have identified and explained many other magnetic states, both uniaxial and two-dimensionally periodic. We find that besides the commonly proposed non-collinear magnetic states also collinear two-dimensional magnetic order can arise due to higher-order interactions. Such collinear multi-Q states can easily be mistaken as skyrmion lattices if one is unaware of the effects discussed in our paper. Our work further demonstrates that the magnetic phase space is significantly enhanced due to the collinear multi-Q states which are verified experimentally in a real material system.

iii. The authors called the collinear multi-Q states a new class of magnetic order. But it looks similar to the so-called magnetic bubbles. Is the difference only in the size?

As the reviewer correctly states the size of typical magnetic bubbles is orders of magnitudes larger than the multi-Q states investigated here. More importantly, the stabilization mechanism has totally different physical origins: whereas magnetic bubbles are stabilized by long-range dipolar interactions, the collinear multi-Q states discovered in our work arise due to higher-order exchange interactions.

iv. It is not clear whether the nano-scale collinear states exhibit interesting transport properties. It is well known that noncollinear states like skyrmions lead to intriguing transport, such as the topological Hall effect, by the Berry phase mechanism. In the collinear case, the Berry phase does not play a role. What kind of interesting transport phenomena do the authors expect for the collinear states?

*We agree with the reviewer that the topological Hall effect should be absent in perfectly collinear multi-Q states. However, these states can be seen as nanoscale ferrimagnetic states since the magnetic moment per unit cell does not vanish. It amounts to about $0.8 \mu_B/\text{Fe atom}$ ($15 \mu_B$ per unit cell) for the 7:12-MS state and about $0.3 \mu_B/\text{Fe atom}$ ($9 \mu_B$ per unit cell) for the 12:15-MS state. Recently, large anomalous Hall effects have been found for antiferromagnets (see e.g. *Nature Reviews Materials* **7**, 482 (2022)) and ferrimagnets (see e.g. *Phys. Rev. Res.* **4**, 013215 (2022)) which make such materials interesting for applications in antiferromagnetic spintronics. The anomalous Hall effect in antiferromagnets and ferrimagnets can also be explained based on the Berry phase concept and it can be calculated using density functional theory. Therefore, we believe that the novel magnetic states found in our work will trigger theoretical work on the transport properties of such states which may also appear e.g. in 2D magnets since these can also exhibit significant higher-order exchange interactions.*

In our revised paper, we have included the references on the anomalous Hall effect in antiferromagnets and ferrimagnets in the conclusion. We have also explicitly given the magnetic moment per unit cell in the collinear mosaic states and skyrmion lattice states in the methods section.

v. The emphasis is on the three-site four spin interaction, but it appears that the biquadratic interaction with $B_1 > 0$ also plays an important role in stabilizing the collinear states. In some previous studies, the biquadratic interaction with $B_1 < 0$ was argued for noncollinear spin textures. The authors are asked to comment on the importance of the positive biquadratic interaction and the reason for the different sign of B_1 .

The reviewer is certainly right that the biquadratic interaction with a positive sign, as in our system, will also prefer collinear over non-collinear spin states. However, due to its large value the three-site four spin term plays the dominant role in our system. In addition, its energy contribution depends much more sensitively on the dimensionality of the spin structures (uniaxial or two-dimensional) than the biquadratic term (cf. green and red curves in lower panels of Fig. 6 in the revised manuscript).

REVIEWERS' COMMENTS

Reviewer #1 (Remarks to the Author):

The authors have made substantive revisions and additions to the manuscript in response to the three reviews. In particular, the clarified discussions of collinear vs. noncollinear textures, and the added structures in Figures 2,3 as well as the simulations in the supplement make the central focus of collinear vs. noncollinear structures much more clear. In addition, I understand the authors' reluctance to quantify the SP-STM contrast in the new supplemental figure with in/out of plane tip contrast, but the additional data does do a good job of illustrating the predominately out-of-plane nature of the spin textures.

This is a substantial body of work, and while I recommend publication, the manuscript still presents as two papers stapled together. In particular, the segue between the experimental and theoretical parts could be strengthened. The paragraph beginning on line 145 of pg 7 is more typical of a concluding paragraph in a paper; instead the paragraph should more succinctly identify the key conclusions from the experiments at a high level, and stress the key remaining questions which can then be addressed with the theoretical calculations. This will greatly help the broad readership of Nature Communications appreciate the work.

Reviewer #2 (Remarks to the Author):

In the revised manuscript, the authors appropriately addressed all the issues raised in the previous communication. I have no further comment, and recommend the publication of the manuscript in Nature Communications.

Reviewer #3 (Remarks to the Author):

I have read carefully all the correspondences and the revised manuscript. I acknowledge that the authors have generally provided adequate responses to my questions. Upon the motivation of this study, I now understand that their interest has changed and expanded from the original motivation through the research, but I still think that it would be better to clarify the previous unresolved

problems and the motivation for this study in the introduction part. I am not fully confident that the discovery of the multi-Q collinear states is of general interest and worthy of publishing in Nature Communications, but if the other two referees recognize the importance of the present work, I will not strongly object to its publication.

REVIEWERS' COMMENTS

Reviewer #1 (Remarks to the Author):

The authors have made substantive revisions and additions to the manuscript in response to the three reviews. In particular, the clarified discussions of collinear vs. noncollinear textures, and the added structures in Figures 2,3 as well as the simulations in the supplement make the central focus of collinear vs. noncollinear structures much more clear. In addition, I understand the authors' reluctance to quantify the SP-STM contrast in the new supplemental figure with in/out of plane tip contrast, but the additional data does do a good job of illustrating the predominately out-of-plane nature of the spin textures.

This is a substantial body of work, and while I recommend publication, the manuscript still presents as two papers stapled together. In particular, the segue between the experimental and theoretical parts could be strengthened. The paragraph beginning on line 145 of pg 7 is more typical of a concluding paragraph in a paper; instead the paragraph should more succinctly identify the key conclusions from the experiments at a high level, and stress the key remaining questions which can then be addressed with the theoretical calculations. This will greatly help the broad readership of Nature Communications appreciate the work.

We thank the reviewer for again assessing our manuscript. We have followed the remaining suggestion of the reviewer and rephrased the last paragraph of the experimental results section to improve the segue between the experimental and theoretical part (as indicated by red colored text in the revised manuscript).

Reviewer #2 (Remarks to the Author):

In the revised manuscript, the authors appropriately addressed all the issues raised in the previous communication. I have no further comment, and recommend the publication of the manuscript in Nature Communications.

We thank the reviewer for taking the time and for the positive recommendation.

Reviewer #3 (Remarks to the Author):

I have read carefully all the correspondences and the revised manuscript. I acknowledge that the authors have generally provided adequate responses to my questions. Upon the motivation of this study, I now understand that their interest has changed and expanded from the original motivation through the research, but I still think that it would be better to clarify the previous unresolved problems and the motivation for this study in the introductory part. I am not fully confident that the discovery of the multi-Q collinear states is of general interest and worthy of publishing in Nature Communications, but if the other two referees recognize the importance of the present work, I will not strongly object to its publication.

We have picked up the concern of this reviewer and emphasized the difference between previous work and our new findings in the introductory part of the manuscript (as highlighted by red text in the revised version of the manuscript).